# PoSE: Efficient Context Window Extension of LLMs via Positional Skip-wise Training

**Dawei Zhu** [*♡♠]    **Nan Yang** [◇]    **Liang Wang** [◇]    **Yifan Song** [♡♠]    **Wenhao Wu** [♡♠]
**Furu Wei** [◇]    **Sujian Li** [♡♠]
[♡] School of Computer Science, Peking University
[♠] National Key Laboratory for Multimedia Information Processing, Peking University
[◇] Microsoft Corporation
https://github.com/dwzhu-pku/PoSE

## Abstract

Large Language Models (LLMs) are trained with a pre-defined context length, restricting their use in scenarios requiring long inputs. Previous efforts for adapting LLMs to a longer length usually requires fine-tuning with this target length (*Full-length* fine-tuning), suffering intensive training cost. To decouple train length from target length for efficient context window extension, we propose **Po**sitional **S**kip-wis**E** (PoSE) training that smartly simulates long inputs using a fixed context window. This is achieved by first dividing the original context window into several chunks, then designing distinct *skipping bias terms* to manipulate the position indices of each chunk. These bias terms and the lengths of each chunk are altered for every training example, allowing the model to adapt to all positions within target length. Experimental results show that PoSE greatly reduces memory and time overhead compared with Full-length fine-tuning, with minimal impact on performance. Leveraging this advantage, we have successfully extended the LLaMA model to 128k tokens using a 2k training context window. Furthermore, we empirically confirm that PoSE is compatible with all RoPE-based LLMs and position interpolation strategies. Notably, our method can potentially support infinite length, limited only by memory usage in inference. With ongoing progress for efficient inference, we believe PoSE can further scale the context window beyond 128k.

## 1 Introduction

Large Language Models (LLMs) have revolutionized language modeling and demonstrated impressive abilities to perform various tasks (Brown et al., 2020). However, even with their remarkable capacity, these LLMs remain restricted by pre-defined *context window* sizes, suffering from notable performance decline when input tokens exceeds these limits. Nevertheless, numerous application scenarios demand extremely long input sequences, including long document summarization (Huang et al., 2021), in-context learning with numerous examples (Li et al., 2023), and long document retrieval (Zhou et al., 2022), etc. This naturally poses a significant challenge of **context window extension**: Extending the context window of a pre-trained LLM to accommodate longer sequences.

Naively fine-tuning LLMs on inputs of target length for window extension has received limited success due to the large disruption introduced by new position indices (Chen et al., 2023a; Han et al., 2023). Addressing this, Position Interpolation (Chen et al., 2023a; kaiokendev, 2023; Peng et al., 2023) propose to down-scale the position indices to match the original window size, yielding improved results for context extension. However, these methods still rely on *Full-length* fine-tuning, i.e., fine-tuning with context of target length, which is memory and time-intensive due to the computational complexity that increases quadratically with input length. For example, Chen et al. (2023a) use 32 A100 GPUs to extend LLaMA models from 2k to 8k context, and 128 A100 GPUs for even larger context. These overhead has made it impossible to extend context window to extreme lengths.

---

[*]Work done during Dawei's internship at MSRA. Sujian Li is the corresponding author.

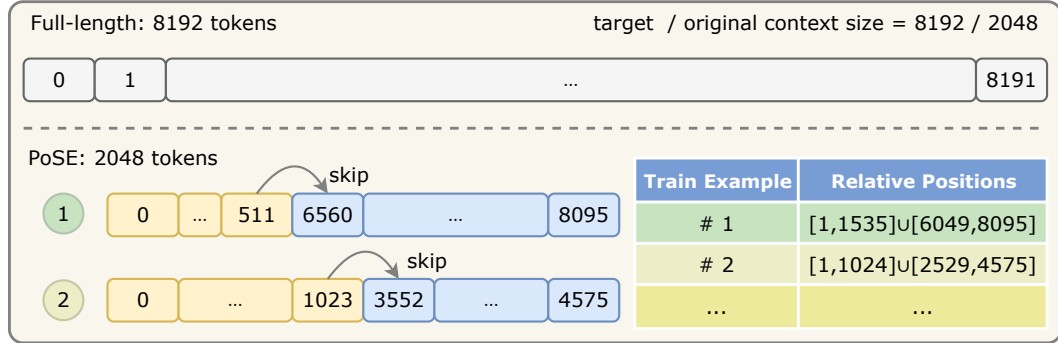

Figure 1: Position indices of Full-length fine-tuning v.s. PoSE fine-tuning for extending the context window size from 2,048 to 8,192. At each iteration, the former directly takes 8,192 tokens for fine-tuning, while PoSE manipulates the position indices of 2,048 tokens to simulate longer inputs. For example, we partition the original context window of 2,048 tokens into two chunks, and adjust the position indices of the second chunk by adding a distinct skipping bias term. These bias terms, as well as the length of each chunk, are altered for each training example, so that the model can adapt to all relative positions of the target context window through fine-tuning.

In this paper, we introduce **Po**sitional **S**kip-wis**E** (PoSE) fine-tuning to decouple the fine-tuning length from the target context window length, unleashing the possibility of efficiently extending context window to an extreme size. The key idea of PoSE is to simulate long inputs by manipulating position indices within a fixed context window. As depicted in Figure 1, we partition the original context window into several chunks, and adjust the position indices of each chunk by adding a distinct skipping bias term. These bias terms, as well as the length of each chunk, are altered for each training example, so that the model can adapt to all positions (including both absolute and relative) within the target context window through fine-tuning. Meanwhile, by maintaining continuous position indices within each chunk, PoSE bears a close resemblance to pre-training. As a result, the model's pre-trained capacity for language modeling and comprehension is retained to the greatest degree.

The advantages of our PoSE are threefold: **1) Memory and Time Efficiency:** By only requiring the original context size for fine-tuning, PoSE circumvents the quadratic increase in computational complexity with respect to target length during the fine-tuning stage, thereby significantly reducing memory and time overhead. 2) **Potential for Extremely-Long Context:** We manage to extend the context window of LLaMA (Touvron et al., 2023a) by up to 64 times (2k→128k, k=1,024) while preserving decent ability of language modeling and understanding. 3) **Compatible with all RoPE-based LLMs and PI strategies:** The effectiveness of PoSE has been empirically validated across several representative RoPE-based LLMs, including LLaMA, LLaMA2 (Touvron et al., 2023b), GPT-J (Wang & Komatsuzaki, 2021), and Baichuan (Baichuan, 2023). Additionally, PoSE has been demonstrated to be compatible with a variety of position interpolation methods, including Linear (Chen et al., 2023a), NTK (Peng & Quesnelle, 2023), and YaRN (Peng et al., 2023) interpolation.

Notably, by decoupling the fine-tuning and target length, PoSE can theoretically extend context window to an infinite length. The only constraint is the memory usage during the inference phase. Hopefully, with the continuous advancements in efficient inference techniques, including Flash Attention (Dao et al., 2022; Dao, 2023), xFormers (Lefaudeux et al., 2022), vLLM (Kwon et al., 2023), etc, we believe PoSE can promisingly push the context window size to a even larger scale.

## 2 RELATED WORK

**Training Length-Extrapolatable Models.** Length extrapolation requires the model to handle continually increasing input tokens, even beyond the context window size used for training (Press et al., 2021). To this end, a series of positional embedding schemes have been proposed, including ALibi (Press et al., 2021), xPos (Sun et al., 2023), NoPos (Haviv et al., 2022), etc.

Similar to our work, Ruoss et al. (2023) also attempted to simulate longer sequences during training time to mitigate out-of-distribution lengths. They proposed randomized positional encoding

(RandPos), which randomly selected an ordered subset of position indices from longer sequences. Our proposed method, PoSE, diverges from their approach in several key aspects: First, RandPos is a positional embedding scheme designed to pre-train encoder-only models from scratch for length extrapolation. In contrast, PoSE is a fine-tuning method aiming at efficiently extend the context window of pre-trained LLMs, which are majorly decoder-only models. Second, in RandPos, the position indices between adjacent tokens are not continuous. However, in PoSE, the position indices within each chunk are intentionally made continuous to resemble the pre-training phase, therefore reducing the risk of disrupting the language modeling abilities learned during pre-training.

**Fine-tuning LLMs for Longer Context.**   Differing from length extrapolation, which primarily involves training a model from scratch to support lengths exceeding those it was initially trained for, context window extension focuses on extending the context window of a pre-trained LLM. Directly fine-tuning an existing LLM with a longer context window has been shown to progress slowly (Chen et al., 2023a). To expedite and stabilize training, Chen et al. (2023a) first down-scaled position indices to match original context size through Linear Position Interpolation. Subsequently, a range of Positional Interpolation (PI) strategies have been introduced, including NTK (Peng & Quesnelle, 2023) and YaRN (Peng et al., 2023). More recently, LongLora (Chen et al., 2023b) propose shift short attention to approximate full attention. However, all these methods require Full-length fine-tuning, suffering computational cost that grows with target context size. By contrast, our method managed to decouple train / target length, requiring only the original context size for fine-tuning.

**Memory Transformers.**   An alternative strategy for extremely long input sequences involves memory mechanisms. Typically, there are two lines of research for utilizing memory: the recurrence-based approach (Dai et al., 2019; Bulatov et al., 2022) and the retrieval-based approach (Wu et al., 2022; Wang et al., 2023; Tworkowski et al., 2023). The former segments long inputs and reuses the hidden states of preceding segments as memory, suffering from information loss and limited capacity for random access. The latter encodes prior sequences as (key, value) pairs and utilizes a memory retriever and reader to extract previously encoded information, primarily limited by the lack of interaction between discrete memory segments. More recently, Mohtashami & Jaggi (2023) introduced landmark attention to facilitates random access to any chunk of the input. In contrast, our method achieves full access to the entire input without any modifications to the attention mechanism.

## 3  METHODOLOGY

### 3.1  PRELIMINARIES

**Rotary Position Embedding (RoPE).**   The use of RoPE (Su et al., 2021) has become pervasive in contemporary LLMs, including LLaMA (Touvron et al., 2023a), GPT-J (Wang & Komatsuzaki, 2021), etc. It encodes position information of tokens with a rotation matrix that naturally incorporates explicit relative position dependency. To elucidate, given a hidden vector $\boldsymbol{h} = [h_0, h_1, ..., h_{d-1}]$, where $d$ is the hidden dimension, and a position index $m$, RoPE operates as follows:

$$
f(\boldsymbol{h}, m) =
\begin{pmatrix} h_0 \\ h_1 \\ h_2 \\ h_3 \\ \vdots \\ h_{d-2} \\ h_{d-1} \end{pmatrix}
\otimes
\begin{pmatrix} \cos m\theta_0 \\ \cos m\theta_0 \\ \cos m\theta_1 \\ \cos m\theta_1 \\ \vdots \\ \cos m\theta_{d/2-1} \\ \cos m\theta_{d/2-1} \end{pmatrix}
+
\begin{pmatrix} -h_1 \\ h_0 \\ -h_3 \\ h_2 \\ \vdots \\ -h_{d-1} \\ h_{d-2} \end{pmatrix}
\otimes
\begin{pmatrix} \sin m\theta_0 \\ \sin m\theta_0 \\ \sin m\theta_1 \\ \sin m\theta_1 \\ \vdots \\ \sin m\theta_{d/2-1} \\ \sin m\theta_{d/2-1} \end{pmatrix}
\tag{1}
$$

where $\theta_j = 10000^{-2j/d}, j \in \{0, 1, ..., d/2 - 1\}$. Unlike previous absolute position encodings that are directly applied to the input vector $\boldsymbol{x}$, RoPE is employed on the query and key vectors at each layer. Given a query $\boldsymbol{q}$ at position $m$ and a key $\boldsymbol{k}$ at position $n$, attention score $a(\boldsymbol{q}, \boldsymbol{k})$ is defined as:

$$
\begin{aligned}
a(\boldsymbol{q}, \boldsymbol{k}) &= < f(\boldsymbol{q}, m), f(\boldsymbol{k}, n) > \\
&= \sum_{j=0}^{d/2-1} [(q_{2j}k_{2j} + q_{2j+1}k_{2j+1}) \cos{(m-n)\theta_j} + (q_{2j}k_{2j+1} - q_{2j+1}k_{2j}) \sin{(m-n)\theta_j}] \\
&:= g(\boldsymbol{q}, \boldsymbol{k}, \boldsymbol{\theta}, m - n)
\end{aligned}
\tag{2}
$$

Hence, RoPE encodes position information in a relative manner, as the attention score depends on the relative distances between positions rather than their absolute position values.

**Problem Formulation.** Given a Large Language Model pre-trained with a context window size of $L_c$, our objective is to extend this context size to a target length $L_t$, so that the model maintains good performance when processing input sequences containing a maximum of $L_t$ tokens.

**Position Interpolation (PI).** In contrast to directly extending the position indices to $L_t - 1$ when dealing with an input text $\boldsymbol{x} = \{x_0, x_1, ..., x_{L_t}\}$, position interpolation down-scales the position indices to align with the original context window size $L_c$. This approach effectively mitigates the risk of encountering extreme values and has been empirically demonstrated to enhance stability during fine-tuning. Various interpolation strategies have been proposed, with $\alpha = L_t/L_c$ denoting the scaling factor:

- *Linear Interpolation.* As described by Chen et al. (2023a) and kaiokendev (2023), linear interpolation involves a proportional down-scaling of the position index $m$ to $m/\alpha$. Consequently, the attention score between a query $\boldsymbol{q}$ at position $m$ and a key $\boldsymbol{k}$ at position $n$ becomes $g(\boldsymbol{q}, \boldsymbol{k}, \boldsymbol{\theta}, (m-n)/\alpha)$, as defined in Equation 2. Theoretical analysis has substantiated that the interpolated attention score exhibits significantly greater stability compared to the extrapolated counterpart.

- *Neural Tangent Kernel (NTK) Interpolation.* In contrast to linear interpolation, NTK Interpolation alters the base of RoPE, effectively modifying the rotational "speed" of each dimension of RoPE (Peng & Quesnelle, 2023). Specifically, the original $\theta_j = 10000^{-2j/d}, j \in \{0, 1, ..., d/2-1\}$ in RoPE is transformed into $\theta'_j = (10000\lambda)^{-2j/d}$, where $\lambda = \alpha^{d/d-2}$. It is noteworthy that the value of $\lambda$ is chosen to ensure that $m\theta'_{d/2-1} = (m/\alpha)\theta_{d/2-1}$.

- *YaRN Interpolation.* Different from Linear and NTK interpolation that treat each dimension of RoPE equally, YaRN (Peng et al., 2023) employs a ramp function to combine Linear and NTK interpolation at varying proportions across different dimensions. Simultaneously, it introduces a temperature factor to mitigate distribution shift of attention matrix caused by long inputs.

### 3.2 Proposed Approach: Positional Skip-wise Training (PoSE)

Although position interpolation effectively addresses out-of-distribution position indices, extending to an extreme length by fine-tuning on context window of this size remains impractical, owing to the quadratic growth in computational complexity of attention as sequence length increases. Instead, we explore to train within the original context window $L_c$ and achieve context window extension via manipulating position indices to simulate longer inputs.

There are two designing desiderata for this endeavor: First, to avoid out-of-distribution positions during inference, the relative distance of manipulated position indices should comprehensively cover the range of $\{1, \ldots, L_t - 1\}$. Second, fine-tuning with the manipulated position indices should not harm the original abilities of LLMs, so the structure of manipulated position indices should closely adhere to the original structure to the greatest extent possible.

Initially, we randomly divide the original context window $L_c$ into $N$ chunks $c_0, c_1, \ldots, c_{N-1}$, each with lengths $l_0, l_1, \ldots, l_{N-1}$, where $\sum_{i=0}^{N-1} l_i = L_c$. We introduce the starting index $st_i$ for each chunk $c_i$, which facilitates the formulation of its position indices as follows:

$$\text{Pos}(c_i) = \{st_i, st_i + 1, \ldots, st_i + l_i - 1\}, \quad st_i = \sum_{j=0}^{i-1} l_j \tag{3}$$

Subsequently, we employ the discrete uniform distribution $\mathcal{U}(S)$ to sample a *skipping bias* term $u_i \sim \mathcal{U}(\{u_{i-1}, \ldots, L_t - L_c\})$ for each chunk $c_i$. This bias term is applied to the corresponding chunk to transform the original position indices into:

$$\text{PoSE}(c_i) = \{u_i + st_i, u_i + st_i + 1, \ldots, u_i + st_i + l_i - 1\} \tag{4}$$

Note that the constraint of $u_i \geq u_{i-1}$ is applied to prevent position index overlaps between chunks.

Intuitively, the introduction of skipping bias terms exposes model to a more diverse range of relative positions. To achieve comprehensive coverage of the target context window, we re-sample both the

length and skipping bias term of every chunk for each training example. Moreover, the continuity of position indices within each chunk closely resembles the structure employed during pre-training. Consequently, fine-tuning the model on these new position indices for language modeling does not compromise its original capabilities.

Concerning the text contained within each chunk, a similar procedure is followed to select continuous spans of tokens from the input text $\boldsymbol{x} = \{x_0, x_1, ..., x_{L_x}\}$. To elaborate, we begin by sampling a bias term $v_i \sim \mathcal{U}(\{v_{i-1}, ..., L_x - L_c\})$ followed by assigning the content of chunk $c_i$ as below:

$$c_i = \boldsymbol{x}[v_i + st_i : v_i + st_i + l_i] \tag{5}$$

Notably, we have also explored other assigning strategy of $v_i$, including scenarios where $v_i = 0$, which results in genuinely continuous content for the chunks, or $v_i = u_i$, aligning the manipulated position indices with actual positions in the original text. However, we observe that these variations have relatively little impact on the outcomes of fine-tuning.

After position indices and content for each chunk are settled, we perform position interpolation for stabilized fine-tuning. For simplicity, We set the initial bias terms $u_0$ and $v_0$ to 0. In terms of chunk number $N$, we view it as an trade-off between efficiency and effectiveness. Because an increase in the number of chunks will further deviates from the position structure of pre-training, which may harm the ability acquired during pre-training. Hence, in this paper we set $N$ to 2, exposing the models to a wider range of relative positions, while adhering as close to the original position structure as possible. (See Appendxi A and B for further discussion of $v_i$ and $N$.)

## 4 EXPERIMENTS

In this section, we conduct experiments to verify the effectiveness of PoSE for context window extension. Our method demonstrates impressive results on context lengths of both 16k and 32k for language modeling as well as passkey retrieval. Other advantages of PoSE are discussed in Section 5.

### 4.1 SETUPS

**Training Procedure.**    For each setting in the main experiments, we train LLaMA-7B with the next token prediction objective. This training process comprises 1,000 steps, employing a global batch size of 64 on 8 V100 GPUs using Deepspeed ZeRO stage 3 (Rajbhandari et al., 2020). We use learning rate $2e^{-5}$ and a linear scheduler, with 10 warmup steps. We use AdamW optimizer with its default hyperparameters setup. The fine-tuning dataset is sourced from The Pile (Gao et al., 2020), with a minimum length requirement of 2,048 tokens. Our default choice for interpolation strategies is linear interpolation. For evaluation, we use a single A100 GPU. Flash Attention V2 (Dao, 2023) is applied, making it possible to evaluate long documents of up to 128k tokens (k=1,024)

**Evaluation Tasks and Datasets.**    We examine the ability of long text modeling on two tasks: language modeling and passkey retrieval. The language modeling task is a fundamental task that reflects the overall capability of a model in handling long text. Passkey retrieval, on the other hand, can effectively measure the maximum distance that a token can attend to during the inference stage. We evaluate language modeling on GovReport (Huang et al., 2021) and Proof-pile (Zhangir et al., 2022) datasets. For passkey retrieval, we follow Mohtashami & Jaggi (2023) to construct synthetic prompts for evaluation.

**Baseline Methods.**    We compare our PoSE training method against following baselines:

- *Full-length* fine-tuning takes input tokens of target length for fine-tuning. For this method, computation complexity scales quadratically with target context window size. Following Chen et al. (2023a) and Peng et al. (2023), we perform PI before fine-tuning LLMs on inputs of target length.

- *RandPos* (Ruoss et al., 2023) is initially designed to train an encoder-only model from scratch for length extrapolation. However, since it shares similar idea of simulating longer sequences via changing position indices, we include it for a comprehensive comparison. Given the original / target context window length $L_c$ / $L_t$, it uniquely samples $L_c$ positions from the set $\{0, ..., L_t - 1\}$, arranges them in ascending order, and employs them as new position indices for training. For fair comparison, we also apply PI for this method.

Table 1: Perplexity of models trained with different methods. We conduct evaluation on the GovReport and Proof-pile datasets, varying evaluation context window size from 2k to 32k. Our PoSE, with a fixed training window size of 2k, effectively extended to a target context size of 16k / 32k for inference while receiving only minimal performance degradation compared to Full-length.

| Method | Context size Train / Target | GovReport | | | | | Proof-pile | | | | |
|---|---|---|---|---|---|---|---|---|---|---|---|
| | | 2k | 4k | 8k | 16k | 32k | 2k | 4k | 8k | 16k | 32k |
| Original | - / - | 4.74 | $>10^3$ | $>10^3$ | $>10^3$ | $>10^3$ | 2.83 | $>10^3$ | $>10^3$ | $>10^3$ | $>10^3$ |
| Full-length | 16k / 16k | 4.87 | 4.70 | 4.61 | 4.59 | - | 2.93 | 2.71 | 2.58 | 2.53 | - |
| RandPos | 2k / 16k | 11.63 | 11.17 | 11.54 | 15.16 | - | 7.26 | 6.83 | 6.76 | 7.73 | - |
| | 2k / 32k | 93.43 | 95.85 | 91.79 | 93.22 | 97.57 | 60.74 | 63.54 | 60.56 | 63.15 | 66.47 |
| PoSE (Ours) | 2k / 16k | 4.84 | 4.68 | 4.60 | 4.60 | - | 2.95 | 2.74 | 2.61 | 2.60 | - |
| | 2k / 32k | 4.91 | 4.76 | 4.68 | 4.64 | 4.66 | 3.01 | 2.78 | 2.66 | 2.60 | 2.59 |

## 4.2 LANGUAGE MODELING

First, we investigate the impacts of different fine-tuning methods on long sequence language modeling using the GovReport and Proof-pile datasets. GovReport is a summarization dataset comprising 19,402 reports published by the Congress and the U.S. Government, with an average document length of 7,866 tokens. We randomly select 50 reports containing more than 32,768 tokens for evaluation. Similarly, Proof-pile is a 13GB mathematical dataset of long mathematical documents. In line with the approach taken for GovReport, we choose 50 samples from Proof-pile that contain more than 32,768 tokens for evaluation.

Table 1 presents the results of scaling to 16k and 32k using Full-length, RandPos, and PoSE training method, each with linear interpolation (See Appendix C for results of NTK and YaRN) . For each scaled model, as well as the *Original* LLaMA model, we report perplexity scores at various evaluation context window sizes, ranging from 2k to 32k, employing the sliding window approach proposed by Press et al. (2021). For evaluation efficiency, we set the stride of the sliding window to 1,024.

First, we observe an overall decreasing trend of perplexity for both models scaled to 16k and 32k via PoSE as evaluation context window size increases, proving their abilities to leverage longer context. Second, with significantly shorter context length during fine-tuning, our PoSE achieves comparable results with Full-length, consolidating its effectiveness. Third, our method achieves much stronger results than RandPos. We suppose it is because our manipulated position indices closely resembles that of pre-training, hereby preserving the pre-trained language modeling ability to the greatest extent.

We also notice that all the scaling methods suffers certain performance degradation as the supported context length increases. We perceive this as a trade-off between the quantity of tokens the model can process and the level of granularity in the attention the model can pay to each individual token.

## 4.3 PASSKEY RETRIEVAL FOR EFFECTIVE CONTEXT WINDOW

To effectively measure the maximum distance that a token can attend to during the inference stage, we adopt the passkey retrieval test proposed by Mohtashami & Jaggi (2023). In this test, models are tasked with recovering a random passkey hidden within a lengthy document. Prompt template used for this task is presented in Figure 2a.

Specifically, we compare the original LLaMA model with the PoSE-extended versions for 16k and 32k context. For each model, we vary the prompt length from 2k to 32k. For each length, we conduct the passkey retrieval test for 50 times, with a random passkey of 5 digits generated and placed at a random position inside the prompt. We also include results from Full-length, RandPos, and PI-only (position interpolation without fine-tuning). Figure 2b illustrates the results. For the Original, PI-only, and RandPos models, their retrieval accuracy rapidly drop to 0 when the context exceeds 2k. In contrast, both PoSE-16k / 32k models managed to maintain a high retrieval accuracy ($\geq 90\%$) within their respective target context window, comparable to Full-length. This indicates that models trained via PoSE genuinely possess the capability to attend to all tokens within the extended context windows.

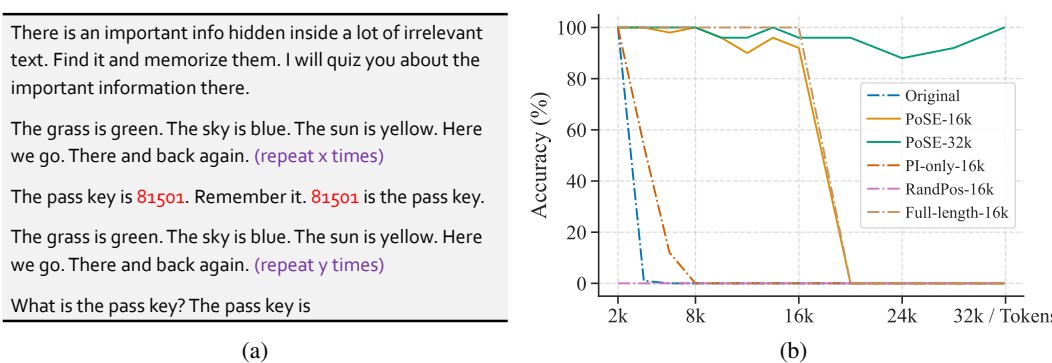

Figure 2: (a) Prompt template used for passkey retrieval; (b) Retrieval accuracy for the PoSE-extended 16k / 32k models, compared with other baselines. Both PoSE-extended models maintain a high retrieval accuracy ($\geq 90\%$) within their respective context window.

## 5 ANALYSIS

In this section, we analyze the advantages of PoSE, including 1) memory and time efficiency; 2) compatibility with all RoPE-based LLMs and diverse interpolation strategies; 3) potential for extremely-long context. In Section 5.4, We also verify that model performance within the original context window only receives minimal degradation.

### 5.1 MEMORY AND TIME EFFICIENCY

We study the memory and time efficiency of PoSE compared with Full-length fine-tuning. For each method, we scale LLaMA-7B to 4k / 8k / 16k through 1,000 training steps with a global batch size of 16 on 8 V100 GPUs. Experiment results are demonstrated in Figure 3. Figure 3(a) and (b) respectively illustrates memory and time consumption for 1,000 steps of Full-length versus PoSE. While the training cost of Full-length increases rapidly with target window length, PoSE only requires a fixed quota of memory and time for context extension, which is significantly lower. Figure 3(c) further compares model perplexity of the two training methods at different steps on GovReport. Notably, both models achieve relatively low perplexity levels within the initial 100 training steps. Moreover, at each step, our proposed PoSE, while requiring only a training context size of 2k tokens, exhibits very close language modeling ability to Full-length fine-tuning, which requires an extended training context of 16k. We did not experiment with context window of 32k or above, because V100 machines cannot afford full fine-tuning of these lengths. But it can be expected that the overhead ration between Full-leng and PoSE will become more exaggerated as target length increases. Consequently, we can confidently assert that our proposed approach is both memory and time-efficient.

### 5.2 COMPATIBILITY WITH ROPE-BASED LLMs AND DIVERSE INTERPOLATION STRATEGIES

We also delve into the effectiveness of PoSE when applied to different RoPE-based LLMs, as well as various interpolation strategies. Specifically, we employ PoSE on four distinct models: LLaMA-7B, LLaMA2-7B, GPT-J-6B, and Baichuan2-7B, all of which encompasses RoPE in their architectures. The original context size of LLaMA-7B and GPT-J-6B is 2k, while that of LLaMA2-7B and Baichuan2-7B is 4k. For each model, we examine the integration with Linear, NTK, and YaRN interpolation, as well as the original version for comparative purposes. The same GovReport dataset as described in Section 4.2 is utilized. The test set is truncated to the first 1k to 16k tokens for plotting the perplexity curve, as depicted in Figure 4. First, it is evident that PoSE is effective across all four models and three interpolation strategies, as evidenced by the low perplexities achieved by all 12 combinations in comparison to the 4 original model. Second, we observe that NTK and YaRN interpolation generally yields superior results compared to Linear interpolation. However, it is noteworthy that NTK exhibits a significant increase in perplexity after a certain turning point, which occurs prior to reaching the target context length. This behavior is consistent with previous findings, indicating that for a given scaling factor $\alpha$, NTK cannot genuinely expand the context window by $\alpha$ times (Peng & Quesnelle, 2023; Quesnelle, 2023; Peng et al., 2023).

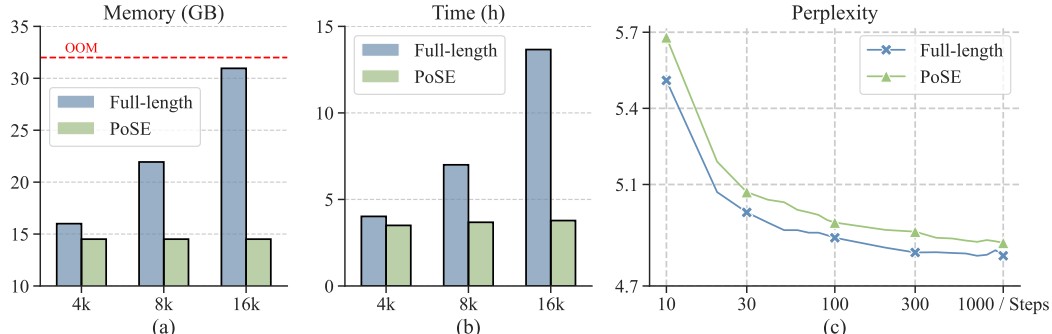

Figure 3: Full-length fine-tuning v.s. PoSE in terms of (a) Memory and (b) Time consumption for extending LLaMA-7B from 2k to 4k / 8k / 16k context, each finishing 1000 training steps. (c) Perplexity of both 16k-context models at every training steps. We show that PoSE takes a constantly reduced time and memory for context extension, while attaining a comparable level of PPL performance with Full-length fine-tuning at each step.

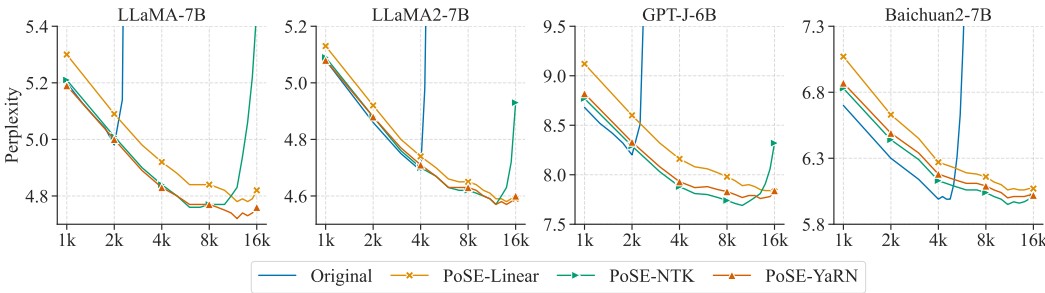

Figure 4: Perplexity of LLaMA-7B, LLaMA2-7B, GPT-J-6B, Baichuan2-7B extended to 16k via PoSE with Linear / NTK / YaRN interpolation, along with the *Original* model. The consistently low perplexity observed across all nine combinations serves as an indication of the effectiveness of our method across RoPE-based LLMs and diverse interpolation strategies.

## 5.3 POTENTIAL FOR EXTREMELY-LONG CONTEXT

Because PoSE only takes a fixed context window at training stage to extend to target context window size, we can promisingly extend LLMs to support infinite input lengths using this method. In this section, we extend context window size to 96k and 128k to explore PoSE's potential for extreme context window extension. Given the need to evaluate on extremely long documents, we have opted to employ two book datasets, namely Books3 (Presser, 2020) and Gutenberg (PG-19) (Rae et al., 2019). Both of these datasets consist of extensive collections of literary works, rendering them well-suited subjects for the assessment of long-range modeling. For our evaluation, we randomly selected 20 books from each dataset, each containing more than 128k tokens.

Fine-tuning LLaMA models using PoSE, we experimented with Linear / NTK / YaRN interpolation for both the 96k and 128k models. To calculate perplexity, we adhere to the sliding window strategy adopted in Section 4.2, with an increased sliding window step of 16k to enhance evaluation efficiency. The outcomes of these experiments are detailed in Table 2. It is observe that, PoSE successfully extends the model's context window to 96k when coupled with Linear interpolation, and further extends the context window to 128k when paired with YaRN. These promising results consolidates the effectiveness of PoSE for extreme context window extension.

## 5.4 EVALUATION OF CAPABILITY ON ORIGINAL CONTEXT WINDOW

In this section, we examine the capabilities of the PoSE-extended models on the original context window using standard benchmarks. We combine the Hugging Face Open LLM Leaderboard (Face, 2023)

Table 2: Perplexity of models extended to extreme context size via PoSE on PG-19 and Books3. We show that our training method can effectively extend context window size to 128k when combined with YaRN interpolation.

| Model | Gutenberg (PG-19) | | | | Books3 | | | |
|---|---|---|---|---|---|---|---|---|
| | 32k | 64k | 96k | 128k | 32k | 64k | 96k | 128k |
| PoSE-Linear-96k | 10.18 | 11.11 | 13.57 | - | 9.98 | 10.90 | 13.42 | - |
| PoSE-NTK-96k | 7.98 | 20.39 | 38.73 | - | 8.29 | 20.82 | 40.39 | - |
| PoSE-YaRN-96k | 8.31 | 8.65 | 9.36 | - | 8.90 | 9.40 | 10.38 | - |
| PoSE-Linear-128k | 16.90 | 22.47 | 26.77 | 31.18 | 26.20 | 43.62 | 57.08 | 70.87 |
| PoSE-NTK-128k | 8.04 | 14.84 | 29.48 | 34.80 | 8.34 | 16.04 | 31.42 | 37.00 |
| PoSE-YaRN-128k | 9.32 | 10.36 | 10.77 | 11.33 | 10.56 | 12.30 | 13.07 | 13.81 |

Table 3: Performance of PoSE-extended LLaMA model on standard benchmarks in comparison with Full-length fine-tuning and the original LLaMA. We show that PoSE-extended models exhibit only marginal performance degradation compared with Full-length fine-tuning and the original version.

| Model | Zero-Shot | | | | Few-Shot | |
|---|---|---|---|---|---|---|
| | BoolQ | PIQA | WinoGrande | TruthfulQA | ARC-C | HellaSwag |
| Original LLaMA | 75.11 | 78.67 | 69.85 | 34.08 | 51.19 | 77.75 |
| Full-Linear-16k | 70.95 | 77.64 | 69.06 | 31.89 | 48.55 | 74.19 |
| Full-NTK-16k | 75.80 | 78.08 | 68.98 | 33.83 | 48.81 | 76.57 |
| Full-YaRN-16k | 73.88 | 77.64 | 68.15 | 34.12 | 50.60 | 77.18 |
| PoSE-Linear-16k | 74.50 | 78.13 | 68.59 | 32.05 | 48.29 | 75.56 |
| PoSE-NTK-16k | 74.28 | 78.24 | 68.90 | 33.89 | 49.83 | 76.82 |
| PoSE-YaRN-16k | 74.28 | 78.02 | 69.06 | 34.00 | 49.23 | 77.04 |
| PoSE-Linear-128k | 67.71 | 76.22 | 67.56 | 36.16 | 39.93 | 66.04 |
| PoSE-NTK-128k | 75.35 | 78.18 | 68.98 | 32.71 | 49.66 | 76.19 |
| PoSE-YaRN-128k | 73.61 | 77.80 | 70.01 | 34.47 | 48.46 | 75.54 |

with a subset of LLaMA benchmarks to assess zero-shot and few-shot performance. For zero-shot evaluation, we employ BoolQ (Clark et al., 2019), PIQA (Bisk et al., 2020), WinoGrande (Keisuke et al., 2019), and TruthfulQA (Lin et al., 2022). For few-shot evaluation, we utilize 25-shot ARC-Challenge (Clark et al., 2018) and 10-shot HellaSwag (Zellers et al., 2019). Our evaluation metrics are benchmark-specific: for BoolQ, PIQA, and WinoGrande, we report accuracy; for TruthfulQA, we report mc2; and for ARC-C and HellaSwag, we report normalized accuracy.

Table 3 summarizes the results. It is observed that, PoSE-extended models exhibit only marginal performance degradation compared with Full-length fine-tuning and the original LLaMA, with the only exception of the 128k model employing linear interpolation. This indicates that while extending context window size, PoSE effectively preserves original language comprehension ability.

## 6 CONCLUSION

In this paper, we introduce **Po**sitional **S**kip-wis**E** (PoSE) training to efficiently extend the context window of Large Language Models. PoSE simulates long inputs by manipulating position indices, thereby requiring only the original context window for fine-tuning, successfully decoupling train length and target length. Experiments have shown that, compared with fine-tuning on the full length, PoSE greatly reduces memory and time overhead. Taking advantage of this, we have managed to extend LLaMA model to 128k on 8 V100 GPUs, observing only minimal performance degradation on standard benchmarks. We have also empirically verified that PoSE is compatible with all RoPE-based LLMs and position interpolation strategies.

## 7 ACKNOWLEDGEMENT

We thank all the anonymous reviewers for their helpful comments on this paper. We thank Xueguang Ma, Yang Ouyang, Pengyun Yue, Hanyu Li, Fangwei Zhu for the thoughtful discussion. This work was partially supported by the Okawa Research Grant.

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

Table 4: Comparison of different methods for choosing $v_i$. We report perplexity with evaluation context window ranging from 2k to 16k. We show that these variations have relatively little impact on the outcomes of fine-tuning.

| Method | GovReport | | | | Proof-pile | | | |
|---|---|---|---|---|---|---|---|---|
| | 2k | 4k | 8k | 16k | 2k | 4k | 8k | 16k |
| $v_i \sim \mathcal{U}(\dots)$ | 4.84 | 4.68 | 4.60 | 4.60 | 2.95 | 2.74 | 2.61 | 2.60 |
| $v_i = 0$ | 4.85 | 4.72 | 4.64 | 4.68 | 2.96 | 2.75 | 2.63 | 2.61 |
| $v_i = u_i$ | 4.84 | 4.68 | 4.60 | 4.60 | 2.95 | 2.73 | 2.60 | 2.56 |

## A   ABLATION OF TEXT CONTAINED WITHIN EACH CHUNK

PoSE divide the original context window into several chunks, and modify the position indices of each chunk to cover a wider range of relative positions in a fixed window. However, it does not impose a particular constraint on the text contained within each chunk. Recall that in Equation 5, we assign the content of chunk $c_i$ as below:

$$c_i = \boldsymbol{x}[v_i + st_i : v_i + st_i + l_i]$$

In this section, we explore several strategies for determining $v_i$: 1) sampling from uniform distribution, $v_i \sim \mathcal{U}(\{v_{i-1}, \dots, L_x - L_c\})$, which is the one used in PoSE; 2) $v_i = 0$, which results in genuinely continuous content for the chunks; 3) $v_i = u_i$, aligning the manipulated position indices with actual positions in the original text. We use the same test setting as Section 4.2, extending LLaMA-7B from 2k to 16k context. As can be seen in Table 4, we show that these variations have relatively little impact on the outcomes of fine-tuning.

## B   ANALYSIS OF CHUNK NUMBER N

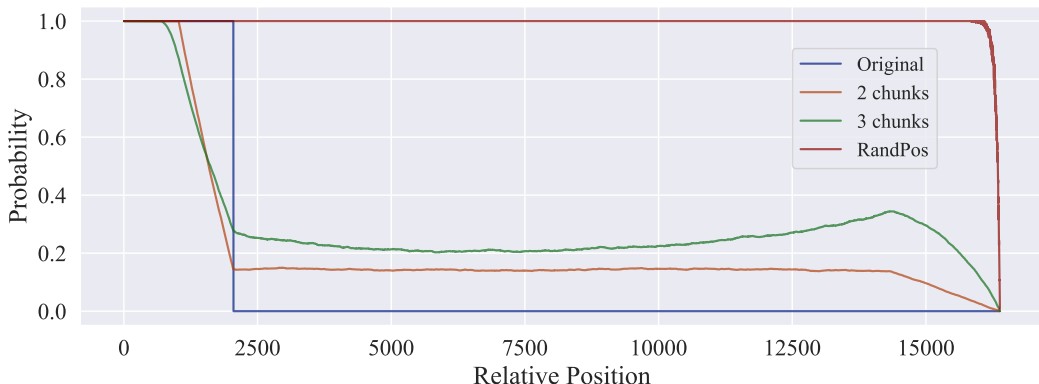

Figure 5: Coverage probability for each relative position in a single training example (2k -> 16k). Utilizing multiple chunks reduces coverage probability within the original $[0, 2,048]$ context window, while enhancing the coverage likelihood of relative positions in the range of $[2,048, 16,383]$. Probability of coverage increases with the number of chunks. Pushing the chunk number to the limit is RandPos, utilizing 2048 chunks, capable of covering every relative position in each training example by expectation.

PoSE achieves coverage of all positions within the target context window by randomly sampling the chunk sizes and skipping bias terms for each training example. In this section, we explore the probability of each relative position being covered by a training example, using a context extension of 2,048 to 16,384 as an example. For the unextended original version, the probability of a relative position within 2048 being covered is 1, and the probability of a relative position above 2,048 being covered is 0. For the cases where the number of chunks is 2, 3, or 2,048 (i.e., RandPos), we use the

```python
visit_prob_list = np.array([0] * Lt)
iter_times = 10000
for _ in range(iter_times):
    l0 = random.randint(1, Lc-1)
    u1 = random.randint(0, Lt-Lc)

    l1 = Lc - l0
    rng1 = set(range(1, max(l0,l1))
    rng2 = set(range(u1+1, u1+))

    rng = rng1 | rng2

    for x in rng:
        visit_prob_list[x] += 1

visit_prob_list /= iter_times
```

```python
visit_prob_list = np.array([0] * Lt)
iter_times = 10000
for _ in range(iter_times):
    l0 = random.randint(1, Lc-2)
    l1 = random.randint(1, Lc-l0-1)
    l2 = Lc - l0 - l1
    u1 = random.randint(0, Lt-Lc)
    u2 = random.randint(u1, Lt-Lc)

    rng1 = set(range(1, max(l0,l1,l2)))
    rng2 = set(range(u1+1, u1+l0+l1))
    rng3 = set(range(u2-u1+1, u2-
u1+l1+l2))
    rng4 = set(range(u2+l1+1, u2+Lc))
    rng = rng1 | rng2 | rng3 | rng4

    for x in rng:
        visit_prob_list[x] += 1

visit_prob_list / = iter_times
```

```python
visit_prob_list = np.array([0] * Lt)
iter_times = 100
for _ in range(iter_times):

    tot_pos_list = list(range(Lt))
    new_pos_list = random.sample(tot_pos_list, Lc)
    new_pos_list.sort()

    distance_rng = set()
    for i in range(0, len(new_pos_list)-1):
        for j in range(i+1, len(new_pos_list)):
            distance_rng.add(new_pos_list[j] -
new_pos_list[i])

    for x in distance_rng:
        visit_prob_list[x] += 1

visit_prob_list /= iter_times
```

Figure 6: Python Code used for calculating coverage probability of each relative position in Figure 5.

Table 5: Comparison of different chunk numbers. We report perplexity with evaluation context window ranging from 2k to 16k. By increasing chunk number, relative positions in $[2,048, 16,383]$ receive an increased chance of being trained, rendering better results for context extension. However, extremely large chunk number also damages model performance.

| Chunk number | Proof-pile | | | |
| --- | --- | --- | --- | --- |
| | **2k** | **4k** | **8k** | **16k** |
| 1 | 2.83 | $> 10^3$ | $> 10^3$ | $> 10^3$ |
| 2 | 2.95 | 2.74 | 2.61 | 2.60 |
| 3 | 2.93 | 2.72 | 2.60 | 2.59 |
| 2048 | 7.26 | 6.83 | 6.76 | 7.73 |

Monte Carlo method to estimate this coverage probability. The code used is demonstrated in Figure 6. The estimated results are shown in Figure 5. It can be seen that PoSE reduces the coverage probability of positions within the original context window, while all relative positions in $[2,048, 16,383]$ receives a certain increase in chance of being covered, and the probability of coverage increases as the number of chunks increases. For the case where the number of chunks is equal to 2,048, the probability of each relative position being covered is close to 1. With this observation, we further compare the impact of chunk number on language modeling capability, as presented in Table 5. Increasing chunk number efficiently renders better results for context extension. However, extremely large chunk number also damages model performance, due to the severe deviation from the position encoding structure used in pre-training phase. We believe that the choice of the number of chunks is a trade-off between training efficiency and performance.

## C  SLIDING WINDOW PPL FROM LINEAR / NTK / YARN INTERPOLATION

Evaluation results in Table 1 are based on Linear interpolation. In this section, we comprehensively provide experiment results with all three PI strategies, and compare four scenarios for each: Full-length fine-tuning, PoSE, PI-only, and Original. Note that *PI-only* means we only apply position interpolation without fine-tuning, while *Original* means the original LLaMA model with neither PI nor fine-tuning. For testing data and sliding window stride, we use the same setup used for Table 1. From Table 6, We can see that for all strategies, the performance follows the same trend: Full-length $\approx$ PoSE $>$ PI-only $\gg$ Original. We also notice that the NTK method suffers from a significant increase in ppl at 16k, mainly because NTK cannot effectively extend the context window by a scaling factor $\alpha$ (Peng & Quesnelle, 2023; Quesnelle, 2023; Peng et al., 2023). YaRN alleviates this issue, achieving progressively decreasing ppl as the context window grows.

Table 6: Perplexity of models trained with different methods using Linear / NTK / YaRN interpolation. It is observed that for all interpolation strategies, the performance follows same trend: Full-length $\approx$ PoSE > PI-only $\gg$ Original.

| Method | Context size Train / Target | GovReport | | | | Proof-pile | | | |
|---|---|---|---|---|---|---|---|---|---|
| | | 2k | 4k | 8k | 16k | 2k | 4k | 8k | 16k |
| Original | - / - | 4.74 | $> 10^3$ | $> 10^3$ | $> 10^3$ | 2.83 | $> 10^3$ | $> 10^3$ | $> 10^3$ |
| **Linear Interpolation** | | | | | | | | | |
| PI-only | - / 16k | 43.80 | 43.35 | 45.89 | 54.33 | 25.32 | 24.20 | 24.88 | 29.59 |
| Full-length | 16k / 16k | 4.87 | 4.70 | 4.61 | 4.59 | 2.93 | 2.71 | 2.58 | 2.53 |
| PoSE (Ours) | 2k / 16k | 4.84 | 4.68 | 4.60 | 4.60 | 2.95 | 2.74 | 2.61 | 2.60 |
| **NTK Interpolation** | | | | | | | | | |
| PI-only | - / 16k | 5.62 | 5.61 | 5.80 | 550 | 3.27 | 3.15 | 3.19 | 517 |
| Full-length | 16k / 16k | 4.78 | 4.63 | 4.57 | 7.24 | 2.93 | 2.71 | 2.61 | 5.66 |
| PoSE (Ours) | 2k / 16k | 4.79 | 4.63 | 4.57 | 7.24 | 2.92 | 2.71 | 2.60 | 4.37 |
| **YaRN Interpolation** | | | | | | | | | |
| PI-only | - / 16k | 5.57 | 5.51 | 5.57 | 5.83 | 3.17 | 2.97 | 2.87 | 2.89 |
| Full-length | 16k / 16k | 4.78 | 4.62 | 4.54 | 4.53 | 2.90 | 2.68 | 2.56 | 2.52 |
| PoSE (Ours) | 2k / 16k | 4.79 | 4.63 | 4.55 | 4.55 | 2.91 | 2.69 | 2.57 | 2.53 |

