# OpenReview forum: "PoSE: Efficient Context Window Extension of LLMs via Positional Skip-wise Training"
_ICLR.cc/2024/Conference — ICLR 2024 poster_

### Official Review · Reviewer_NXZX · 2023-10-29

**Soundness:** 2 fair
**Presentation:** 2 fair
**Contribution:** 2 fair
**Rating:** 6
**Confidence:** 3

**Summary:**

the paper presents pose, a novel mechanism to extend the context window of LLMs that decouples train length from target length. It achieves this by simulating longer inputs instead of directly training on longer inputs.

**Strengths:**

1. the paper is well written; the method is well illustrated.
2. the problem is important, and the method can greatly reduced the memory and computation requirements.

**Weaknesses:**

1. section 3.1 and 3.2 are better to be placed in a different background section, or in related work.
2. the evaluation for long context is limited - they are mostly all perplexity based. The only available non-perplexity experiment is the one with passkey retrieval, but the baselines are not comprehensive. Can the authors either provide more non-perplexity based measure, or include more baseline for Figure 2?

**Questions:**

Please address the weakness above.

---

> ### Author Response · Authors · 2023-11-17
> **Thanks for your valuable feedback!**
>
> We sincerely appreciate your efforts in reviewing our paper and providing valuable advice. We are pleased that you recognize our paper for addressing a significant problem, substantially reducing memory and computation requirements, and being well-written and illustrated. Regarding the issues you raised, we will reply in detail and hope that our response could address your concerns. Your kind consideration of potentially reevaluating the rating would be greatly valued if our responses address your concerns.
>
> **Q1. Section 3.1 and 3.2 are better to be placed in a different background section, or in related work.**
>
> Thank you for the suggestion. Section 3.1 introduces the background, and we will separate it into a "Preliminaries" section. Section 3.2 introduces our methodology, and we will present it as a standalone section.
>
> **Q2. Limited evaluation for long context. Either provide more non-perplexity based measures or include more baselines in the passkey retrieval test in Figure 2.**
>
> Based on your valuable feedback, we have added some new baselines for the passkey retrieval test. We believe this will make our evaluation for long context more comprehensive.
>
> In the original version of our paper, we use the following models for the passkey retrieval test, as in Figure 2:
>
> - Original: The original LLaMA with neither position interpolation nor fine-tuning
>
> - PoSE-16k: the PoSE-extended 16k model
>
> - PoSE-32k: the PoSE-extended 32k model
>
>
> In the revised version of our paper, we have newly added the baselines below:
>
> - PI-Only-16k: Apply position interpolation for 16k context on LLaMA.
>
> - RandPos-16k: the RandPos-extended 16k model.
>
> - Full-length-16k: the 16k model extended via full-length fine-tuning.
>
>
> The evaluation results have been updated in Figure 2b. We can see that, without surprise, Full-length-16k is the best within 16k context, and PoSE-16k & 32k are very closely behind. On the contrary, PI-only-16k, RandPos-16k and Original perform poorly. We think that these models could provide a comprehensive comparison for passkey retrieval test, and hope to clarify your concerns.

---

> > ### Comment · Reviewer_NXZX · 2023-11-17
> >
> > Thanks a lot for getting back to the review. The reviewer is happy to see the problem being addressed. I think this paper should be accepted at ICLR 2024.

---

### Official Review · Reviewer_VfnN · 2023-10-30

**Soundness:** 3 good
**Presentation:** 3 good
**Contribution:** 3 good
**Rating:** 6
**Confidence:** 4

**Summary:**

In this paper, the authors introduce a novel technique known as Positional Skip-wise Training (PoSE) for extending the context window of Large Language Models (LLMs). While existing methods like Positional Interpolation (PI), YaRN, and NTK Interpolation have proven effective in extending the context window of LLMs, they require extensive fine-tuning of the LLM with the entire context window. This fine-tuning process imposes a high computational cost due to the quadratic complexity of LLMs relative to the length of the context window.

To address this challenge, the authors propose a strategic approach during training by selectively skipping certain positions in the middle for each batch. As illustrated in Figure 1, this technique allows the model to effectively perceive a context length of 8192 tokens while using a training-time context window of just 2048 tokens. Consequently, this method significantly reduces the training cost. When evaluated on prominent long-context datasets, including GovReport, Proof-pile, and PG19, the authors demonstrate that PoSE achieves comparable performance to full-context fine-tuning, all while delivering a substantial speedup (as shown in Figure 3).

**Strengths:**

1. The paper is well-written and offers clear, accessible explanations.

2. The results presented in Table 1 demonstrate the effectiveness of the proposed method in extending the context of Language Model (LLM). Furthermore, the performance of PoSE is comparable to that of full-length fine-tuning, highlighting the evident benefit of skipping contiguous token chunks.

3. PoSE excels in both memory reduction and computation time, as illustrated in Figure 3.

4. PoSE proves to be adaptable across various positional encoding interpolation methods, including linear, NTK, and YaRN, thereby enhancing its versatility and applicability.

**Weaknesses:**

I am uncertain about whether PoSE has genuinely extended the contextual window. While the current PPL evaluation results are undoubtedly promising, there's a possibility that methods demonstrating low PPL on PG19/GovReport might not effectively comprehend longer contexts. It's possible that the proposed method excels in consistently generating fluent text over extended periods but falls short in truly grasping the nuances of extended context. If that is the case, the potential impact of PoSE could be rather limited, since there are training-free methods (StreamingLLM, LM-Infinite) that can achieve this goal.

To address this concern, I recommend that the authors consider conducting additional experiments using benchmarks like LongBench and ZeroSCROLLS. Additionally, employing benchmarks utilized by Llama 2 Long could provide valuable insights into the capabilities of PoSE in handling more extensive contextual information.

**Questions:**

I believe that PoSE demonstrates proficiency in both PPL and key retrieval evaluations. I look forward to your response to my inquiry regarding the real extension of the context window, as mentioned in the Weaknesses section.

In addition, I am curious about the performance of the proposed method when the first trunk does not initiate from position 0. Recent concurrent works, such as LM-Infinite and StreamingLLM, have highlighted the significance of retaining the first few tokens in the KV cache for zero-shot context extension, without requiring fine-tuning. I am intrigued to know if the design of $u_0=v_0=0$, which resembles the approach proposed in these two papers, holds the same relevance when applied to a fine-tuned LLM. It would be greatly appreciated (though not obligatory) if the authors could include a discussion on the efficacy of fine-tuning in comparison to these zero-shot methods. For example, does the PPL on long context benchmarks improve with PoSE? Does PoSE unlock new capabilities which zero-shot methods do not have? I will update the score to 8 if the authors provide interesting discussions.

---

> ### Author Response · Authors · 2023-11-17
> **Thanks for your valuable feedback!**
>
> Thank you very much for your valuable feedback! We sincerely appreciate your recognition of our paper as well-written and offers clear, accessible explanations; our method as effectiveness, time and memory efficient, and adaptable across various position interpolation methods. In response to your inquiries, we provide the following explanations.
>
> **Q1. Whether PoSE has genuinely extended the contextual window.**
>
> **A1.** We really appreciate this concern! It is verfied that PoSE has indeed extended the contextual window, as evidenced by the passkey retrieval test (Figure 2). As you mentioned, a lower PPL does not necessarily mean that the model fully understands the entire long input, as in extreme cases, one can always make the model only focus on the last $L_c$ tokens (where$L_c$ is the original context length of the model) using a sliding window approach, and achieve a low perplexity. However, this is not possible for the passkey retrieval test, which requires the model to truly see all input tokens. Our model achieved very high accuracy (>90%) in the target context window, which is strong evidence that our method can genuinely see full sequence during inference. See **A3** below for more discussion with StreamingLLM and LM-Infinite.
>
>
>
> **Q2. Performance of the proposed method when the first chunk does not initiate from position 0.**
>
> **A2.** Thanks for your question! We would like to clarify this. PoSE's design focuses on relative distance rather than absolute distance, so whether the first chunk starts from position 0 or not does not affect the results. Let $L_c$ and $L_t$ denote the original and target context window lengths, respectively, the length of the first chunk be $l_0$, the bias term be $u_0$, the length of the second chunk be $L_c-l_0$, the bias term be $u_0$, we compare the results of two settings: v1) $u_0=0$, i.e., the first chunk starts from position 0; v2) $u_1=L_t-L_c$, i.e., the second chunk ends at position $L_t-1$. Other settings are same as Table 1. The experimental results are attached below. It can be seen that whether the first chunk starts from position 0 or not does not affect the experimental results.
>
> ||GovReport||||Proof-Pile||||
> |---|---|---|---|---|---|---|---|---|
> ||2k|4k|8k|16k|2k|4k|8k|16k|
> |PoSE-16k-v1|4.84|4.68|4.60|4.60|2.95|2.74|2.61|2.60|
> |PoSE-16k-v2|4.85|4.69|4.60|4.61|2.96|2.74|2.62|2.58|
>
> We really appreciate your association with StreamingLLM. However, although they may seem to have some similarities, they are essentially very different. In Discussion1, we will explain this in detail.

---

> ### Author Response · Authors · 2023-11-17
> **Thanks for your valuable feedback! (cont.)**
>
> **Q3. Performance comparison with zero-shot methods like StreamingLLM and LM-Infinite.**
>
> **A3.** Thank you for this question. We would like to clarify the differences between our approach and StreamingLLM & LM-Infinite in this part. StreamingLLM and LM-Infinite are primarily designed for streaming applications, allowing the model to operate continuously on extremely long inputs without extensive memory or dependency on past data[1]. They do extend the contextual window, and can not see long context by design [1]. Our method, however, genuinely extends the context window, providing several benefits:
>
> **1) Strong Performance in passkey retrieval**. When the input length exceeds $L_c$, zero-shot methods like StreamingLLM and LM-Infinite experience a significant decrease in accuracy on the passkey retrieval task [2]. In contrast, our model achieved very high accuracy (>90%) in the target context window.
>
> **2) Lower PPL for long context**. By extending context window, models trained with PoSE can better model long-term dependency, gaining lower PPL for long context. For this, we sampled test data exceeding 16k from the GovReport & Proof-Pile dataset and calculated the average PPL of the first 2k, 4k, 8k, and 16k tokens using PoSE-16k, Full-Length-16k, and StreamingLLM, respectively. To maximize the ability of StreamingLLM, we set the cache size to the pre-training window length (i.e., 2048), instead of the default 256, to allow it to retain as much context information as possible. However, this also greatly reduced the running speed of StreamingLLM, so for each dataset, we sampled 10 texts. The experimental results are shown in the following table:
>
> ||GovReport||||Proof-Pile||||
> |---|---|---|---|---|---|---|---|---|
> ||First 2k|First 4k|First 8k|First 16k|First 2k|First 4k|First 8k|First 16k|
> |Full-Length-16k|4.91|4.60|4.44|4.42|4.33|3.64|2.91|2.65|
> |PoSE-16k|4.87|4.57|4.43|4.42|4.32|3.65|2.93|2.68|
> |StreamingLLM|4.76|4.59|4.61|4.70|4.08|3.60|3.05|2.93|
>
> As shown, the performance of PoSE and Full-Length is very close, indicating that PoSE can achieve close effect as full-length fine-tuning. For the first 2k tokens, StreamingLLM achieved slightly better performance than PoSE, which is reasonable as its effect is equivalent to the original LLaMA at this stage. However, as the input becomes longer, especially for 16k context, PoSE largely outperforms StreamingLLM. Hence, we believe that PoSE is superior to zero-shot methods in long text modeling.
>
>
>
> **Discussion1. Relevance with the initial tokens of KV cache used in StreamingLLM.**
>
> **A.** We are very happy to share with you our observations, although it may not be directly related to our paper. The initial tokens used in StreamingLLM **do not refer to tokens with position ids starting from 0, but rather those at the beginning of the sequence**, which only attend to a few or no tokens. For reasons that may include serving as an Attention Sink [3], the hidden states of these initial tokens are crucial for the proper functioning of the model, while the hidden states of subsequent tokens do not have this effect. This phenomenon holds for absolute position encoding, relative position encoding, even NoPE. In a caching scenario, if the initial tokens are evicted, the model cannot function properly. In contrast, in a non-caching scenario, the hidden states of initial tokens are always recalculated and thus possess the aforementioned property. PoSE does not evict initial tokens, and there is no need to force $u_0=0$. Based on this dicussion, we can see that StreamingLLM and PoSE are orthogonal work. We hope this answers your question and welcome further discussion.
>
>
>
> **References**
>
> [1] The FAQ part of StreamingLLM. https://github.com/mit-han-lab/streaming-llm#faq
>
> [2] Figure 4 of LM-Infinite (version 1). https://arxiv.org/pdf/2308.16137v1.pdf
>
> [3] Efficient Streaming Language Models with Attention Sinks

---

> > ### Comment · Reviewer_VfnN · 2023-11-17
> > **Acknowledgement of author response**
> >
> > Thank you for your detailed author response. I believe that this paper should be accepted to ICLR 2024.

---

### Official Review · Reviewer_T2GT · 2023-11-01

**Soundness:** 3 good
**Presentation:** 3 good
**Contribution:** 3 good
**Rating:** 6
**Confidence:** 4

**Summary:**

The paper proposed Positional Skip-wisE (PoSE) training for extending the context length of LLMs. The idea is to divide the input into multiple chunks and add randomly sampled bias values to later chunks. This effectively simulates the positions the model will see during the inference stage. Experiments with LLaMA show that PoSE is effective in extending RoPE-based LLMs, and is compatible to other length extension techniques like linear, NTK, and YaRN,

**Strengths:**

1. The idea of PoSE is neat and novel. PoSE is able to reduce the GPU memory requirement for training long-context LLMs.
2. Experiments show that PoSE is effective in extending models trained on 2k context length to 32k. Also, the author explored extending models trained on 2k to 128k in Table 2 and demonstrated the potential of PoSE.

**Weaknesses:**

1. Finetuning the model with PoSE will degrade its performance on standard benchmarks, according to Table 3. It is unclear if it can be mitigated from the paper. Also, the perplexity of PoSE is worse than finetuning with the target sequence length (shown in Table 1). Thus, if the user really cares about the performance, finetuning with the target sequence length still gives the best result.
2. From Table 1, I haven't seen the comparison between PoSE and applying NTK directly without training. In addition, the author has not provided enough details in how it conducted full-length finetuning. Is it to first extend the sequence length to 16K via linear or NTK and finetune the model on the 16k sequence length?

**Questions:**

How did you do full-length finetuning? Have you used length expansion before finetuning on the target sequence length?

---

> ### Author Response · Authors · 2023-11-17
> **Thanks for your valuable feedback!**
>
> Thank you for evaluating our method as neat and novel, reducing the memory requirement for context window extension, and demonstrating great potential to extend to 128k. We appreciate your suggestions very much! Below, we will address your questions carefully.
>
> **Q1. Fine-tuning with PoSE degrades performance on standard benchmarks (Table 3).**
>
> **A1.** Thanks for your valuable feedback! Actually, it is common for context-extended models to have a slight performance degradation on short-context tasks compared to the original model. Even GPT-4 has this issue[5]. In Table 3, not only PoSE but also Full-length show a decrease in performance compared to the Original. However, in most cases, the performance degradation is marginal (<5%), which we believe is negligible compared to the benefits of an extended context window.
>
>
>
> **Q2. PoSE is worse than full-length fine-tuning (Table 1). If the user really cares about the performance, finetuning with the target sequence length still gives the best result.**
>
> **A2.** Thanks for your concern! Our work is a trade-off between performance and efficiency. While we acknowledge that full-length fine-tuning yields better results, its training cost is prohibitively high. In comparison, PoSE sacrifices 1% of performance (Table 1) but offers significant memory and time advantages (Figure 3). Therefore, we believe that PoSE is meaningful for most researchers who cannot afford high training budget, as well as large companies aimed at reducing cost for extending to extreme lengths, such as 256k. In either way, PoSE can promisingly promote the development of the long context field.
>
>
>
> **Q3. Add comparison between PoSE and applying NTK directly without training.**
>
> **A3.** We sincerely agree with this and have experimented with all three PI strategies. Since we observed the same trend, we only reported the results of Linear interpolation in Table 1. For Linear / NTK / YaRN interpolation, we compared four scenarios: Full-length, PoSE, PI-only, and Original. _PI-only_ means we only apply position interpolation without fine-tuning. _Original_ means the original LLaMA model with neither PI nor fine-tuning. The table below shows the performance of the 16k models from Linear / NTK / YaRN interpolation, under the same settings as Table 1. We can see that for all strategies, the performance follows the same trend: Full-length ≈ PoSE > PI-only >> Original. We also noticed that the NTK method suffers from a significant increase in ppl at 16k, mainly because NTK cannot effectively extend the context window by a scaling factor $\alpha$  [3]  [4] . YaRN alleviates this issue, achieving progressively decreasing ppl as the context window grows. We appreciate your suggestion and have included these results in Appendix C.
>
>
>
> | Method       | Length |        | GovReport |         |         |         | Proof-Pile |         |         |         |
> | ------------ | ------ | ------ | --------- | ------- | ------- | ------- | ---------- | ------- | ------- | ------- |
> |              | Train  | Scaled | 2k        | 4k      | 8k      | 16k     | 2k         | 4k      | 8k      | 16k     |
> | Original     | -      | -      | 4.74      | >$10^3$ | >$10^3$ | >$10^3$ | 2.83       | >$10^3$ | >$10^3$ | >$10^3$ |
> | **Linear ↓** |        |        |           |         |         |         |            |         |         |         |
> | PI-only      | -      | 16k    | 43.8      | 43.35   | 45.89   | 54.33   | 25.32      | 24.20   | 24.88   | 29.59   |
> | Full-length  | 16k    | 16k    | 4.87      | 4.70    | 4.61    | 4.59    | 2.93       | 2.71    | 2.58    | 2.53    |
> | PoSE         | 2k     | 16k    | 4.84      | 4.68    | 4.60    | 4.60    | 2.95       | 2.74    | 2.61    | 2.60    |
> | **NTK ↓**    |        |        |           |         |         |         |            |         |         |         |
> | PI-only      | -      | 16k    | 5.62      | 5.61    | 5.80    | 550     | 3.27       | 3.15    | 3.19    | 517     |
> | Full-length  | 16k    | 16k    | 4.78      | 4.63    | 4.57    | 7.24    | 2.93       | 2.71    | 2.61    | 5.66    |
> | PoSE         | 2k     | 16k    | 4.79      | 4.63    | 4.57    | 7.24    | 2.92       | 2.71    | 2.6     | 4.37    |
> | **YaRN ↓**   |        |        |           |         |         |         |            |         |         |         |
> | PI-only      | /      | 16k    | 5.57      | 5.51    | 5.57    | 5.83    | 3.17       | 2.97    | 2.87    | 2.89    |
> | Full-length  | 16k    | 16k    | 4.78      | 4.62    | 4.54    | 4.53    | 2.9        | 2.68    | 2.56    | 2.52    |
> | PoSE         | 2k     | 16k    | 4.79      | 4.63    | 4.55    | 4.55    | 2.91       | 2.69    | 2.57    | 2.53    |
> |              |        |        |           |         |         |         |            |         |         |         |

---

> ### Author Response · Authors · 2023-11-17
> **Thanks for your valuable feedback! (cont.)**
>
> **Q4. How is full-length fine-tuning conducted.**
>
> **A4.** In our paper, full-length fine-tuning is conducted through interpolation followed by training, which is a routine approach adopted in the PI [1] and YaRN [2] papers. This approach has shown to have the best performance compared to not fine-tuning or not interpolating. For instance, when extending the window size from 2k to 16k, we first use Linear, NTK, or YaRN interpolation to extend the sequence length to 16k, and then fine-tune the model on the 16k sequence length. The same interpolation strategy is also applied to RandPos for fair comparison. Thank you for your suggestion, we have made this clearer in the `Baseline Methods` part of Section 4.1
>
>
>
> **Reference**
>
> [1] Extending Context Window of Large Language Models via Positional Interpolation
>
> [2] YaRN: Efficient Context Window Extension of Large Language Models
>
> [3] (reddit) NTK-Aware Scaled RoPE allows LLaMA models to have extended (8k+) context size without any fine-tuning and minimal perplexity degradation
>
> [4] (reddit) Dynamically Scaled RoPE further increases performance of long context LLaMA with zero fine-tuning
>
> [5] LooGLE: Can Long-Context Language Models Understand Long Contexts? (Table 5)

---

### Official Review · Reviewer_UVnU · 2023-11-01

**Soundness:** 4 excellent
**Presentation:** 3 good
**Contribution:** 3 good
**Rating:** 6
**Confidence:** 5

**Summary:**

The paper proposes a method called Positional Skip-wise Training (PoSE) to extend the context window size of large language models (LLMs) during fine-tuning. PoSE simulates long inputs by manipulating position indices within a fixed training context window, decoupling the training length from the target length.

**Strengths:**

1. The proposed PoSE approach is effective in extending the context window size of various RoPE-based LLMs to up to 128k tokens with minimal performance degradation compared to full-length fine-tuning.

2. PoSE significantly reduces the memory and time complexity compared to full-length fine-tuning, achieving up to 64x speedup.

3. PoSE is compatible with various RoPE-based language models and positional interpolation strategies.

4. The idea of manipulating position indices within chunks to simulate longer inputs is novel and effective as shown in the experimental results.

**Weaknesses:**

1. The theoretical analysis of PoSE is limited. More discussion of how manipulating position indices allows the model to generalize to longer contexts could strengthen the paper.

2. The experimental setups and hyperparameters are not thoroughly described. More details would increase reproducibility.

3. It would be better that if the authors could provide some failure cases that this method might leads to some missing details or inferior answers to the standard fine-tuned models. Because it should be inevitable that the speedup has a cost on performance. Only the perplexity comparison is a bit unclear for understanding. It would be more clear that the perplexity gap lies in retrieval, summarization, or some other abilities.

**Questions:**

Please see the weakness.

---

> ### Author Response · Authors · 2023-11-17
> **Thanks for your valuable feedback!**
>
> Thank you for recognizing our idea as simple and effective, greatly reducing memory and time overheads, and being adaptable to various RoPE-based models and position interpolation strategies. We really appreciate your valuable feedback. In terms of your concerns, we have addressed them carefully below:
>
> **Q1: Limited theoretical analysis. Add more discussion of how manipulating position indices allows the model to generalize to longer contexts.**
>
> We genuinely appreciate the suggestion to enhance the theoretical analysis in our paper. So far, we have provided some empirical analysis in Appendix B. Based on this, we can potentially explain by analyzing the probability of each relative position being covered by a training example. Let $L_c$ and $L_t$ denote the original and target context window lengths, respectively. For full-length fine-tuning, the probability of a relative position within $L_c$ being covered is 1, and the probability of a relative position in $[L_c, L_t-1]$ being covered is 0. PoSE reduces the coverage probability of positions in $[0,L_c-1]$, while all relative positions in $[L_c,L_t-1]$ receive a certain increase in chance of being covered, making it possible to simulate long inputs with a short context window.
>
> Another potential explanation is the Conceptual Model for Relative Position Attention recently proposed by LM-Infinite[1]. This theory suggests that for a long context, the beginning is responsible for encoding absolute position information, the ending tokens are helpful for the normal operation of the attention layer, and the middle tokens contain relatively less position information. This explains why we can skip middle tokens while receiving only marginal performance drop.
>
> **Q2: More details on experimental setups and hyperparameters**
>
> Thank you for your suggestion. We use learning rate $2e^{-5}$and a linear scheduler, with 10 warmup steps. We use AdamW optimizer with its default hyperparameters setup. We have added this information to the `Training Procedure` in Section 4.1.
>
> **Q3: Provide some failure cases that this method might leads to some missing details or inferior answers to the standard fine-tuned models.**
>
> Compared to Full-length fine-tuning, PoSE achieves very close performance in terms of both perplexity and passkey retrieval accuracy. Inspired by your valuable suggestions, we have identified some instances where full-length outperforms PoSE, but we have also found cases where PoSE outperforms full-length. Therefore, we believe that the differences observed in these individual cases do not imply systematic distinctions between the two approaches. We would like to develop more rigorous methods to analyze the difference between PoSE and Full-length fine-tuning in the future.
>
> **Reference**
>
> [1] LM-Infinite: Simple On-the-Fly Length Generalization for Large Language Models

---

> > ### Comment · Reviewer_UVnU · 2023-11-23
> > **Reply to the response by the authors**
> >
> > Thanks for your detailed clarification and response. I will keep my rate for this paper.

---

### Meta-Review · Area_Chair_uaGi · 2023-12-05

**Metareview:**

This paper introduces Positional Skip-wisE training (PoSE), a new approach for efficiently extending the context window of large language models beyond their initial training limits. The core idea is to simulate long inputs within a fixed context window by dividing the context into chunks and applying distinct skipping bias terms to manipulate their position indices, effectively decoupling the training length from the context length. The method shows promising results in extending the context window size up to 128k tokens with minimal performance degradation.

After the discussion phase, the reviewers unanimously supported the acceptance of this paper. The main concerns of the reviewers were the limited theoretical analysis, lack of detail in experimental setups, and limited evaluation for long context. The authors addressed these concerns by providing additional theoretical insights, clarifying the experimental procedures, and presenting experimental results for additional baselines. Overall, the AC is happy to recommend the acceptance.

**Justification For Why Not Higher Score:**

No reviewer strongly supports this paper.

**Justification For Why Not Lower Score:**

All reviewers suggest acceptance.

---

### Decision · Program_Chairs · 2024-01-16

Accept (poster)